# Windows of opportunity for daily physical activity

**Marvin Du** *

College of Chemistry, University of California, Berkeley, CA, United States of America

* marvin.du@berkeley.edu

**Citation:** Du M (2020) Windows of opportunity for daily physical activity. PLoS ONE 15(9): e0238713. https://doi.org/10.1371/journal.pone.0238713

**Data Availability Statement:** All relevant data are available from GitHub (https://github.com/marvin-du/california-air-quality-data).

**Funding:** The author(s) received no specific funding for this work.

## Abstract

Air pollution is a serious concern to people who want to engage in physical activities to improve and maintain their health. However, air quality data collected at the nearest monitoring site may not be the best source of information because the data may be incomplete (e.g., some pollutants are not monitored) or not representative. This paper puts forward a method that uses air quality data from a large area to derive a diurnal profile of air quality variation for that area and identify the time window in which the air quality is typically the best during a day. If people exercise in that time window then they can minimize their exposure to air pollution. Three years' worth of air quality data in five California counties were analyzed to identify the general pattern of diurnal variation of air quality. The analysis shows the pattern of air quality variation is very similar among those five counties which represent diverse geographical and meteorological conditions. The analysis further reveals that in California air quality is generally the best in early mornings; as such, people should exercise in the early morning if their daily schedule allows it. A similar analysis can be performed for other areas to help people choose the best time window to exercise.

## Introduction

Air pollution is a serious concern to the general public, in particular to those who are engaged in physical activities such as cycling, walking, and running, etc. It is well established that physical activity improves people's health; however, at the same time, physical activity increases intake of air pollutants. As such, increased exposure to air pollution can negate some benefits gained from physical activity or even bring harm when air quality (quantified by the concentrations of the criteria pollutants such as $PM_{10}$ and $PM_{2.5}$) is bad and the duration of physical activity is prolonged [1]. A review article [2] presents findings that aerobic exercise increases the overall inhaled air pollution dose, potentiates the diffusion of pollutants into circulating blood, augments oxidative stress and inflammation. In particular, the inhalation of particulate matter during exercise can raise blood pressure, impair vascular function, and unfavorably affect autonomic balance.

A comprehensive study [3] (and reference therein) demonstrates that in general the beneficial effects of physical activity outweigh the detrimental effects associated with the increased intake of air pollutants. A similar conclusion is reached [4] from a systematic review of online

**Competing interests:** The authors have declared that no competing interests exist.

databases that active commuting has a net benefit to offset the increased inhalation dose of fine particulate matter. Nevertheless, numerous studies show that perception of the harm of air pollution discourages people from engaging in physical activity. For example, it is shown [5] that an increased level of air pollution is associated with reduced leisure-time physical activity, particularly among normal-weighted people. It is shown [6] that a high level of $PM_{2.5}$ significantly discourages physical activity among freshmen students in Beijing, China.

It goes without saying that the general public's awareness of the harm associated with air pollution has some effect to discourage people from engaging in physical activity, especially when air quality is known to be poor. If people can find current air quality information published by public health and environment protection government agencies, they can make an informed decision whether physical activity is preferable with the given air quality. For example, during episodes of very bad air quality, people should avoid intense physical activity to minimize their exposure to air pollutants. Government agencies can issue warnings when air quality is bad and recommend physical activity when air quality is predicted to be good. One example of this type of advisory is Government Canada's webpage (https://weather.gc.ca/airquality/pages/index_e.html) where the air quality health index for all cities and towns is constantly updated based on the most recent air quality monitoring data.

However, air quality varies constantly with time and real-time air quality monitoring data may not be always readily available. Arguably California has the most extensive air monitoring network in the world. According to a recent report [7], California has over 250 monitoring sites and more than 700 air monitors; however, there are still many areas that do not have adequate data for people to make an informed decision of whether and when to exercise. For example, six counties (Alpine, Baja, Lassen, Modoc, Sierra and Yuba) do not have any air monitors and in some counties that have air monitors, only one or two pollutants are monitored. For instance, in Amador and Tuolumne Counties, each has just one ozone monitor and all other pollutants are not monitored. Even in populous counties, not all pollutants are monitored. For example, Santa Clara County (with a population of about two million) does not have any $PM_{10}$ monitors. Another issue with air quality monitoring data is their representativeness. For ozone and nitrogen dioxide, data at the monitoring site can be representative of a fairly large area. But for carbon monoxide and sulfur dioxide, it is hardly the case because those pollutants are directly emitted from pollution sources; therefore source distribution has a strong impact on the area of interest (i.e., the area where people are to exercise). Even for particulate matters, direct emissions can contribute significantly thus making the representativeness of the data questionable.

Air quality monitoring networks are designed and deployed to meet the need of air quality standard attainment demonstration. A lack of monitoring sites may then signify overall good air quality, so it is not surprising that the network may not provide adequate information for people to determine the best time to exercise. In areas that are already in attainment of air quality standards, people still prefer to exercise when the air quality is better than other times but the monitoring data may not be available to guide the selection of the best time to exercise. An alternative to directly using air quality monitoring data at the nearest site is to develop a profile of diurnal variation of air quality from data collected in broader data-rich areas. The diurnal pattern can be used to guide people to exercise in a window of time when air quality is the best throughout the day. If the window of time can fit in with people's daily schedules of work and chores, they will more likely develop a routine for physical exercise and adhere to it for a long time. The objective of this paper is to use air quality monitoring data to identify the best window of time for physical exercise.

Air quality data from five data-rich counties in California in a three-year period (2014–2016) are analyzed to develop the diurnal patterns of air quality variations.

## Methods

### Data

The data used in this study are from the California Air Resources Board's Air Quality and Meteorological Information System (AQMIS) (https://www.arb.ca.gov/aqmis2/aqdselect.php?tab=hourly). All pollutants identified by the U.S. EPA as criteria pollutants except lead are included in the study. Lead is not included because the national ambient air quality standard only has a 3-month standard for it. Hourly monitoring data of sulfur dioxide ($SO_2$), nitrogen dioxide ($NO_2$), carbon monoxide (CO), ozone ($O_3$), respirable particulate matter ($PM_{10}$) and fine particulate matter ($PM_{2.5}$) in five counties (Fresno, Los Angeles, Orange, Sacramento, and Santa Clara) for the period of 2014–2016 were downloaded from AQMIS. These populous counties were selected to represent different geographical areas and diverse air quality conditions. Adequate data coverage was also a consideration when selecting counties (some counties in the State do not have data for some pollutants). Table 1 lists number of monitors for each pollutant in each county.

### Index of unhealthiness

An indicator, named index of unhealthiness, is introduced as the measure of how bad the air quality is for a given hour of day. For each pollutant, index of unhealthiness, $I_{pollutant}$, is defined as

$$I_{pollutant}(hour) = \left( \frac{1}{N_s N_y N_m N_d} \sum_{site=1}^{N_s} \sum_{year=1}^{N_y} \sum_{month=1}^{N_m} \sum_{day=1}^{N_d} \frac{c(site, year, month, day, hour)}{S_{pollutant}} \right) \quad (1)$$

where $N_s$ is the number of monitoring sites in a county for which the index is calculated, $N_y$ is the number of years which is 3 in this study because 3 years' worth of data are used, $N_m$ is the number of month in a year and $N_d$ is the number of days in a month taking into account that $N_d$ varies between 28 and 31. $I_{pollutant}(hour)$ is a function of hour which takes the value of the hour. For example, hour = 8 in the period from 8:00 am to 9:00 am. $S_{pollutant}$ is the one-hour national air quality standard. A smaller value of $S_{pollutant}$ signifies a requirement for better air quality, but it is not practically feasible to set it at zero. For $SO_2$, $NO_2$ and CO, one-hour air quality standards do exist; however, for $O_3$, $PM_{2.5}$ and $PM_{10}$, the air quality standards are for multiple-hour averages. For example, the $O_3$ standard is for the 8-hour average concentration. In the present study those multi-hour standards need to be converted to an equivalent one-hour standard because the objective of the present study is to identify the hour(s) that the air quality is the best. With multiple-hour air quality standards the air quality data will have to be averaged over those same hours so that they can be compared to the standard. In this way, it is only possible to determine if 8-hour ozone and 24-hour particulate matter concentrations are good or bad. If the hourly air quality data are compared with the multiple-hour standard, it

**Table 1. Number of monitoring sites for each pollutant in each county.**

| County Name | Number of Monitors | | | | | |
|---|---|---|---|---|---|---|
| | $SO_2$ | $NO_2$ | CO | $O_3$ | $PM_{2.5}$ | $PM_{10}$ |
| Fresno | 1 | 5 | 4 | 5 | 4 | 2 |
| Los Angeles | 4 | 14 | 14 | 13 | 5 | 2 |
| Orange | 1 | 4 | 5 | 4 | 1 | 1 |
| Sacramento | 1 | 7 | 4 | 7 | 5 | 2 |
| Santa Clara | 1 | 2 | 2 | 4 | 3 | 0 |

won't be readily seen how good or bad the air quality really is for that hour. For example, if one-hour $PM_{2.5}$ is compared with the 24-hour standard, it is difficult to quantify how good or bad the air quality (in terms of $PM_{2.5}$) is in that particular hour. Also, for the sake of consistency, when quantifying the state of air quality in a particular hour, hourly air quality data have to be compared against hourly air quality standards.

Because air quality standards are designed for the maximum, or extreme, levels of air pollution, the equivalent one-hour standard can be estimated from the multi-hour standard with the scaling factors that relate the one-hour and multi-hour extreme concentrations [8] with

$$C_{1-hour} = \frac{C_{n-hour}}{F},$$ (2)

where $C_{1-hour}$ is the one hour peak concentration, $C_{n-hour}$ is n-hour peak concentration, and F is a pollutant-specific scaling factor which can never be zero otherwise n-hour peak concentration is identically zero while the 1-hour concentration peak concentration is not. Table 2 lists the scaling factors for $O_3$, $PM_{2.5}$ and $PM_{10}$. For $O_3$, the equivalent one-hour standard is 100 ppb (from the 8-hour standard of 70 ppb and a scaling factor of 0.7), and for $PM_{2.5}$ and $PM_{10}$, the equivalent one-hour standards are 87.5 and 375 $\mu g/m^3$, respectively (calculated from 24-hour standards of 35 and 150 $\mu g/m^3$, and a scaling factor of 0.4). In the present analysis, the one-hour (or equivalent one-hour) standards are taken to be 196 $\mu g/m^3$, 100 $\mu g/m^3$, 40 $mg/m^3$, 100 ppb, 87.5 $\mu g/m^3$, 375 $\mu g/m^3$ for $SO_2$, $NO_2$, CO, $O_3$, $PM_{2.5}$ and $PM_{10}$, respectively. It should be acknowledged that the method of estimating the equivalent one-hour standard is not robust, but for the present study this approach is acceptable because it provides a reasonable method of judging if hourly air quality is good or poor.

For the purpose of selecting a window of time for physical exercise, it is more meaningful to consider the levels of all pollutants, so a combined index of unhealthiness for all six pollutants is defined as

$$I_{all} = \frac{1}{6} \sum_{pollutant=1}^{6} I_{pollutant},$$ (3)

where the subscript 'pollutant' stands for a pollutant among all six pollutants. The hours during which the index of unhealthiness is the lowest are the hours when the air quality is the best so those hours will be the window of time for daily physical exercise.

## Results

The combined index of unhealthiness in five counties, based on three years' worth of air quality data of all criteria pollutants, is shown in Fig 1. It is clear that the index has a substantial diurnal variation and is the lowest in the early morning, suggesting that exercising in the early morning hours will be subjected to the least impact from air pollution. It should be noted that the patterns of diurnal variations in all five counties are very similar, suggesting the temporal variation pattern is likely universal in the entire state.

In the calculation, each pollutant possesses an equal weight in the index, but studies [9] have shown that different pollutants can have different severities of impact on human health.

**Table 2. Scaling factors to relate one-hour and multi-hour peak concentrations.**

| Pollutant | Number of hours | Scaling factor |
|---|---|---|
| $O_3$ | 8 | 0.7 |
| $PM_{2.5}$ | 24 | 0.4 |
| $PM_{10}$ | 24 | 0.4 |

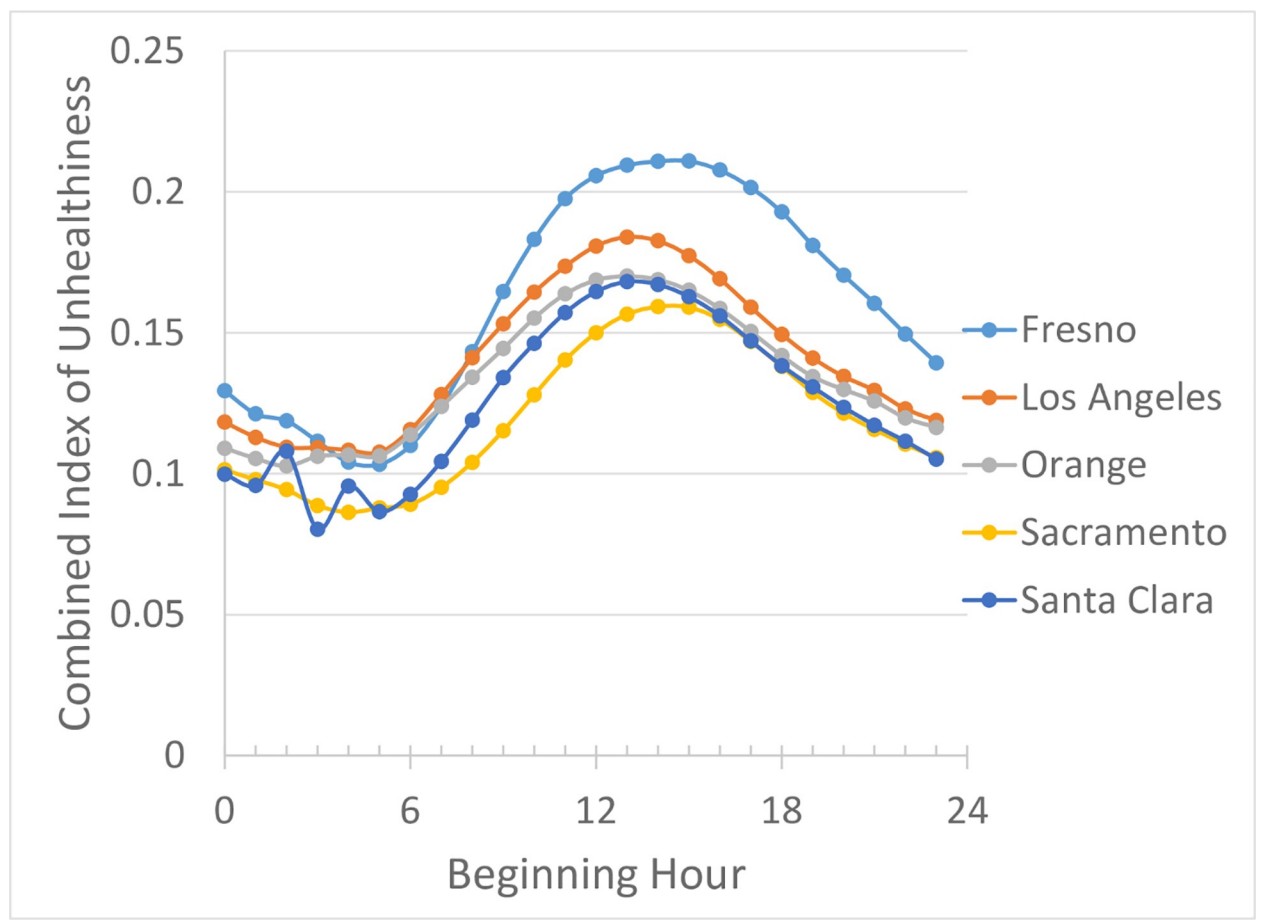

**Fig 1. Diurnal variation of the combined index of unhealthiness for all pollutants in five counties.**

As such, it may be helpful to take a closer look at the index of unhealthiness for each pollutant. Fig 2 shows that the index of unhealthiness for all pollutants is dominated by ozone; and further, only ozone exhibits a strong diurnal variation. The indices of unhealthiness for all other pollutants are relatively flat, including $PM_{2.5}$ which is the main driver of premature mortality [9]. S1 Fig of Supporting Information shows that in each of the five counties the dominating pollutant is ozone, although the degree of dominance by ozone over other pollutants can be different.

The strong diurnal variation of ozone is anticipated because tropospheric ozone is produced only in daytime and the production rate is strongly associated with solar radiation. It is not entirely clear why the other pollutants stay relatively flat beyond the fact the direct emissions contribute to their concentrations, but the consistency of this invariability as demonstrated by the data in all five counties suggests that it probably is universally true as a general trend (albeit not necessarily true for any given location on any given date).

A question the reader may ask is if the diurnal pattern exhibited in Fig 1 is an outcome of the long-term average of air quality data. Or in other words, if averaging across different seasons/months is not performed, do we still obtain similar diurnal variations? If the answer is no, then the best hours for physical exercise may be different for different seasons/months, making the result shown in Fig 1 less useful since different months may have a different hour of day as the optimal time to engage in physical activities. Fig 3 shows that in Los Angeles

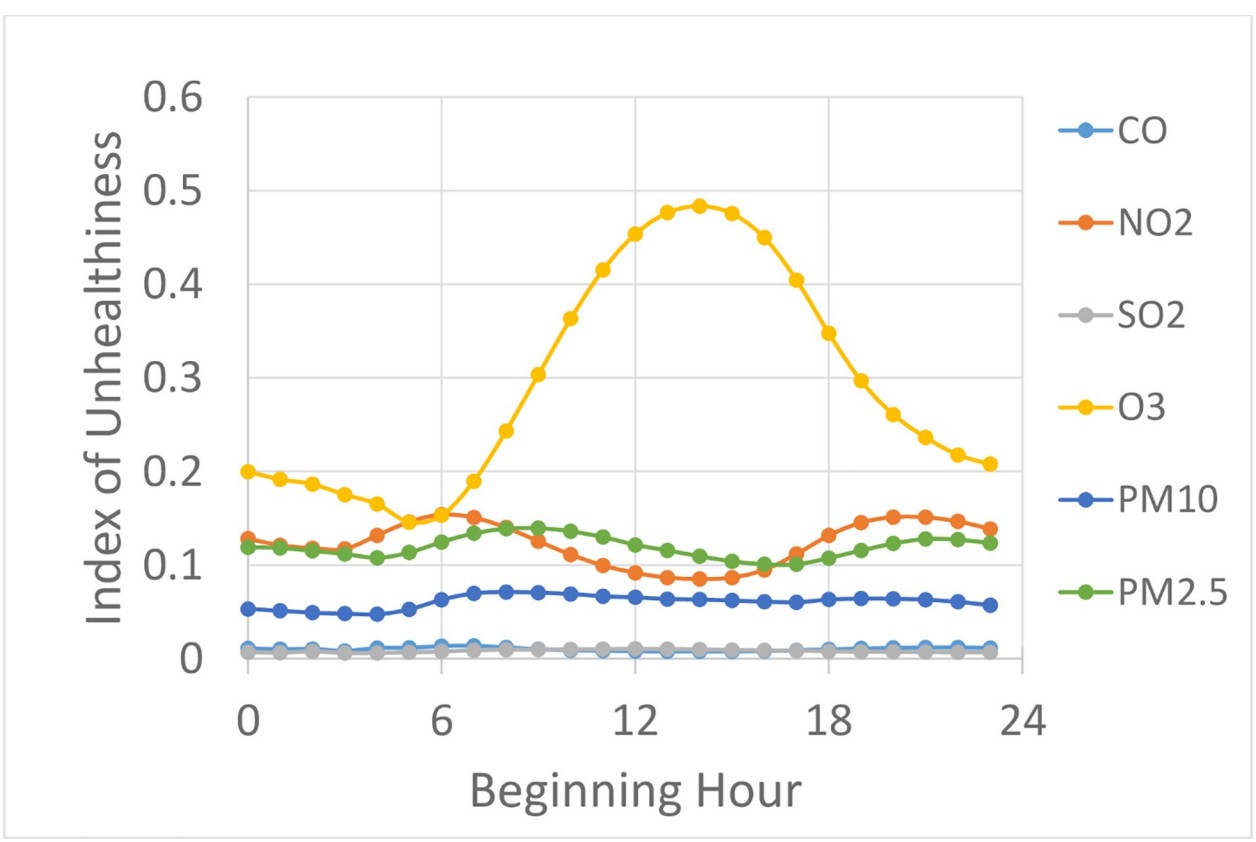

**Fig 2. Diurnal variation of the index of unhealthiness for each individual pollutant.**

County the diurnal variations in different months are very similar, meaning that the optimal window of opportunity for physical activity does not vary with month. Similar results in the other four counties are shown in S2 Fig of the Supporting Information. Together they show that the diurnal pattern in different counties and in different months is the same, and the index of unhealthiness is always the lowest in the early morning.

The work by Giles and Koehle [10] and studies referred to therein recommend exercising in early morning in summer months based on consideration of ozone concentrations. The present study extends that recommendation substantially: exercise in the early morning in all four seasons because air quality, not just the ozone concentrations, is the best in the early morning hours.

## Discussion and conclusion

Based on the analysis of air quality data from multiple years and multiple counties in California, a diurnal pattern of index of unhealthiness is found to be valid in all counties from which air quality data are analyzed, and air quality in the hour from 5 a.m. to 6 a.m. is generally the best. As such, in California the best time for physical activity is that hour. If that one-hour window does not fit a person's daily schedule, it is also acceptable to exercise before 8 a.m. or after 8 p.m.

Strictly speaking, the aforementioned time window may be valid just in California. In other regions the time window for physical activities may be different as the dominating air pollutants may be different. Nevertheless, the method of developing a diurnal pattern of air quality

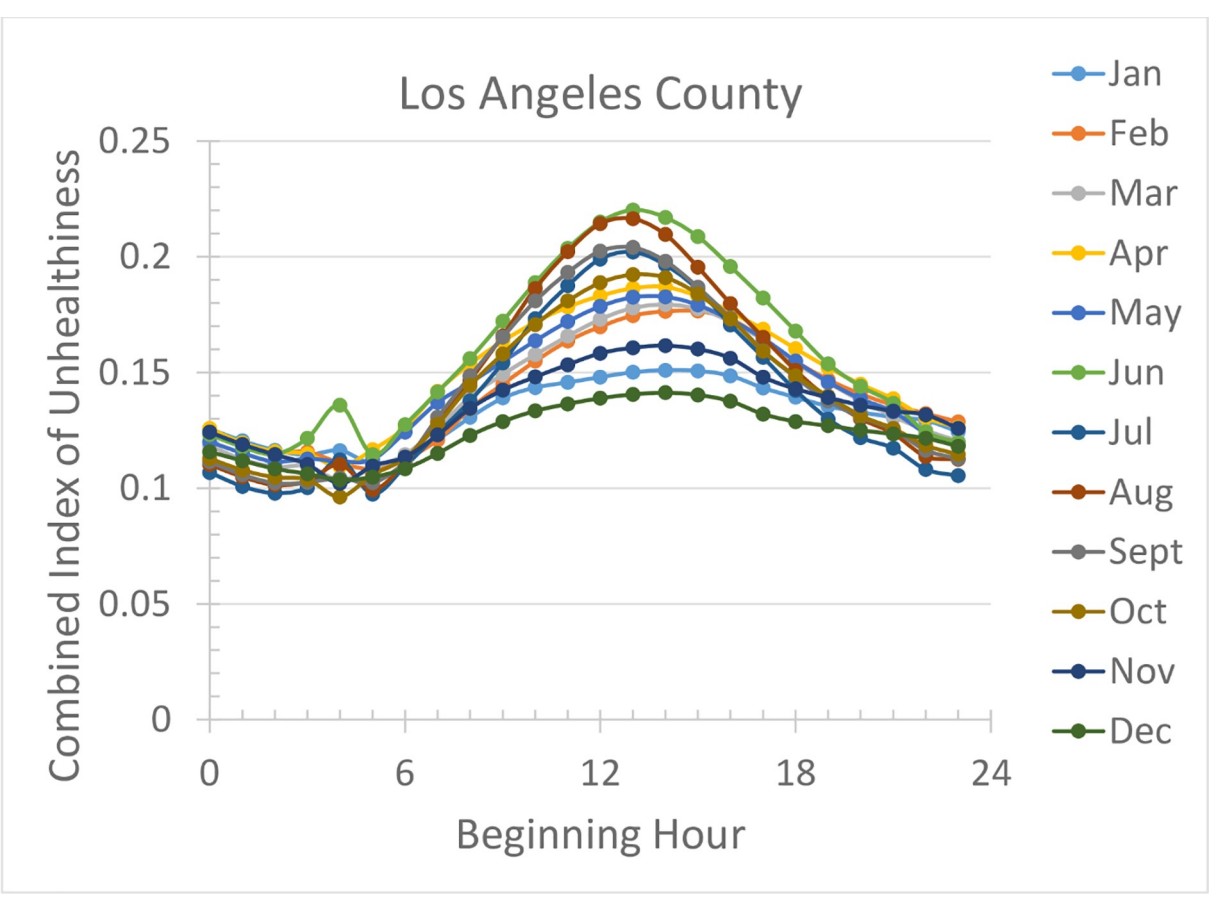

**Fig 3. Diurnal variation of the index of unhealthiness in different months in Los Angeles county.**

variation and to identify the best time window for exercise remains valid. When using air quality data to calculate the index of healthiness for an area, the area should be large enough so that valid data exist for all pollutants at multiple monitoring sites, and the area should also be small enough so that the 'local' characteristic is preserved. It should be noted that these recommendations are based on air quality data averaged over multiple days and a broad area; for any particular day at any particular location, air quality in those hours may not be the best. This 'rule of thumb' should be considered as a general advice when health advice based on representative real-time air quality data is not readily available. For example, if a person is to exercise in an area that is downwind of a monitoring site, and if there are major emissions sources (like a road way, or an industrial/commercial facility) of air pollutants located downwind of the monitoring site but upwind of the exercise area, air quality data at the monitoring site is in no way representative of the exercise area, and the air quality in the area of interest may be much worse than what the monitored air quality data show. Under those circumstances, using the diurnal pattern of air quality variation outlined in this study is a better alternative because the diurnal profile is based on a broad set of data therefore exercising while the index of unhealthiness is the lowest is likely to be better than in other periods of the day.

When an air quality related warning or advice is issued by the authorities, for example, when an area is heavily impacted by wildfires, or when air quality is forecasted to be poor in a broad area, the warning/advice takes precedence.

The objective of this work is to find ways to encourage people to engage in physical activities. There are a number of approaches to promote physical activity. At the level of policy making and municipal planning, building more healthy neighborhoods and improving street connectivity can encourage people to engage physical activities [11–15]. At the personal level, to minimize the harm caused by air pollution, people can work proactively by designing jogging routes to avoid areas of high air pollution. For example, whenever possible, people should exercise in an area away from any known air pollution sources such as industrial facilities, gas stations and major roadways and select a path upwind of those sources of air pollutants. People can also develop a routine for daily exercise based on knowledge of how air quality varies throughout the day. The present analysis provides useful information to address the concerns that people may have about the potential impact of air pollution on exercise. This information can help people develop a routine to exercise in a time window when air quality is generally the best, if their daily schedule allows, and adhere to the routine for a long time. When an air pollution related advisory is not in place, air quality is generally good enough for people to exercise. Our work provides a means to reduce exposure to air pollution and exhort more people to engage in physical activities. For those who are concerned with the impact of air pollution on health benefits gained from physical activities, their stress can be alleviated so that they can be more relaxed and take on more exercises.

The present analysis can also help sport teams to plan their daily training schedule so that players are subjected to the minimum air quality impact.

## Supporting information

**S1 Fig. Index of unhealthiness for all individual pollutants in five California counties.** (a) Fresno, (b) Los Angeles, (c) Orange, (d) Sacramento, and (e) Santa Clara. In all five counties, ozone is the pollutant dominating the overall index of unhealthiness.
(DOCX)

**S2 Fig. The diurnal variations of the combined index of unhealthiness in four other counties.** (a) Fresno, (b) Orange, (c) Sacramento, and (d) Santa Clara. All have similar patterns to the one for Los Angeles County as shown in Fig 3.
(DOCX)

## Author Contributions

**Conceptualization:** Marvin Du.

**Data curation:** Marvin Du.

**Formal analysis:** Marvin Du.

**Investigation:** Marvin Du.

**Methodology:** Marvin Du.

**Project administration:** Marvin Du.

**Resources:** Marvin Du.

**Validation:** Marvin Du.

**Visualization:** Marvin Du.

**Writing – original draft:** Marvin Du.

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
