## [Decision Letter · Decision Letter 0]

3 Aug 2020

PONE-D-20-17624

Windows of Opportunity for Daily Physical Activity

PLOS ONE

Dear Dr. Du,

Thank you for submitting your manuscript to PLOS ONE. After careful consideration, we feel that it has merit but does not fully meet PLOS ONE’s publication criteria as it currently stands. Therefore, we invite you to submit a revised version of the manuscript that addresses the points raised during the review process.

We look forward to receiving your revised manuscript.

Kind regards,

Maria Alessandra Ragusa, PhD Professor

Academic Editor

PLOS ONE

Additional Editor Comments:

Dea coresponding author,

in attached there are the 2 reports.

Please, follow the instructions and upload as soon as possible the revised version.

Afterthat, if all the changes are correctly made, you will receive the final acceptance.

Best regards.

Journal Requirements:

Could you therefore please include the title page into the beginning of your manuscript file itself, listing all authors and affiliations?

4. Please include copies of Tables 1 and 2 which you refer to in your text on pages 5 and 6.

Reviewers' comments:

Reviewer's Responses to Questions

**Comments to the Author**

1. Is the manuscript technically sound, and do the data support the conclusions?

Reviewer #1: Yes

2. Has the statistical analysis been performed appropriately and rigorously? 

Reviewer #1: Yes

3. Have the authors made all data underlying the findings in their manuscript fully available?

Reviewer #1: Yes

4. Is the manuscript presented in an intelligible fashion and written in standard English?

Reviewer #1: Yes

5. Review Comments to the Author

Reviewer #1: see file

6. PLOS authors have the option to publish the peer review history of their article (what does this mean?). If published, this will include your full peer review and any attached files.

Reviewer #1: **Yes: **Abdolrahman Razani

---

## [Author Response · Author response to Decision Letter 0]

12 Aug 2020

All the response is in the file 'Responses to the reviewers.doc'

---

## [Editor Report · Decision Letter 1]

17 Aug 2020

PONE-D-20-17624R1

Windows of Opportunity for Daily Physical Activity

PLOS ONE

Dear Dr. Du,

Thank you for submitting your manuscript to PLOS ONE. After careful consideration, we feel that it has merit but does not fully meet PLOS ONE’s publication criteria as it currently stands. Therefore, we invite you to submit a revised version of the manuscript that addresses the points raised during the review process.

We look forward to receiving your revised manuscript.

Kind regards,

Maria Alessandra Ragusa, PhD Professor

Academic Editor

PLOS ONE

Additional Editor Comments (if provided):

Dear author,

it seems to me that not all the suggestions are followed.

Please read again the referee’s reports and send again the new revised version.

Best regards.

---

## [Author Response · Author response to Decision Letter 1]

19 Aug 2020

Please see the 'Responses to Reviewers' as attached in this submission.

---

## [Editor Report · Decision Letter 2]

24 Aug 2020

Windows of Opportunity for Daily Physical Activity

PONE-D-20-17624R2

Dear Dr. Du,

We’re pleased to inform you that your manuscript has been judged scientifically suitable for publication and will be formally accepted for publication once it meets all outstanding technical requirements.

Kind regards,

Maria Alessandra Ragusa, PhD Professor

Academic Editor

PLOS ONE
---

## [Editor Report · Acceptance letter]

1 Sep 2020

PONE-D-20-17624R2 

Windows of Opportunity for Daily Physical Activity 

Dear Dr. Du:

I'm pleased to inform you that your manuscript has been deemed suitable for publication in PLOS ONE. Congratulations! Your manuscript is now with our production department. 

Kind regards, 

on behalf of

Dr. Maria Alessandra Ragusa 

Academic Editor

PLOS ONE